# Front-Line Tyrosine Kinase Inhibitors in Pediatric Chronic Myeloid Leukemia: A Study on Efficacy and Safety

**DOI:** 10.3390/cancers15153862

**Published:** 2023-07-29

**Authors:** Jae Won Yoo, Suejung Jo, Moon Bae Ahn, Seongkoo Kim, Jae Wook Lee, Myungshin Kim, Bin Cho, Nack-Gyun Chung

**Affiliations:** 1Department of Pediatrics, Seoul St. Mary’s Hospital, College of Medicine, The Catholic University of Korea, Seoul 06591, Republic of Korea; hoiring0209@gmail.com (J.W.Y.); j.crystal1107@gmail.com (S.J.); ksk3497@catholic.ac.kr (S.K.); dashwood@catholic.ac.kr (J.W.L.); chobinkr@catholic.ac.kr (B.C.); 2Catholic Genetic Laboratory Center, Seoul St. Mary’s Hospital, College of Medicine, The Catholic University of Korea, Seoul 06591, Republic of Korea; microkim@catholic.ac.kr; 3Department of Laboratory Medicine, Seoul St. Mary’s Hospital, College of Medicine, The Catholic University of Korea, Seoul 06591, Republic of Korea

**Keywords:** tyrosine kinase inhibitors, treatment response, adverse events, chronic myeloid leukemia, pediatrics

## Abstract

**Simple Summary:**

Tyrosine kinase inhibitors (TKIs) have significantly improved treatment outcomes in pediatric patients with chronic myeloid leukemia (CML). However, there is insufficient evidence suggesting the superiority of one TKI over another in terms of treatment response and long-term adverse events (AEs). We aimed to assess the efficacy and safety profiles of front-line TKIs among pediatric patients. The complete cytogenetic response rates were excellent for both imatinib and dasatinib at 12 months. However, patients treated with dasatinib exhibited significantly faster and higher cumulative rates of early, major, and deep molecular responses. Although both TKIs were well tolerated, a notable decline in height was observed with both TKIs, with a greater decrease observed in the dasatinib group during the last year of observation. Our findings confirm the efficacy of both TKIs; however, the long-term AEs associated with their use should be evaluated in a large cohort of pediatric patients.

**Abstract:**

We conducted a retrospective study on 51 pediatric patients with newly diagnosed chronic myeloid leukemia chronic phase or accelerated phase. The patients were classified into the IMA group (N = 33), treated with imatinib, and the DSA group (N = 18), treated with dasatinib, as front-line tyrosine kinase inhibitors (TKIs). At 12 months, the rates of complete cytogenetic response were similar between the IMA group (92.3%) and DSA group (100%) (*p* = 0.305). However, the rate of early molecular response was higher in the DSA group than in the IMA group (100.0% vs. 80.0%, *p* = 0.043). By 12 and 24 months, the DSA group showed faster and higher cumulative rates of both major (DSA group: 72.2% and 100%, respectively; IMA group: 41.2% and 68.7%, respectively; *p* = 0.002) and deep molecular responses (DSA group: 26.0% and 43.6%, respectively; IMA group: 13.8% and 17.5%, respectively; *p* = 0.004). Both TKIs were well tolerated. Although the height standard deviation scores decreased in both groups, the height decline was greater in the DSA group between one and two years from the start of TKI therapy. In this study, dasatinib achieved faster and higher molecular responses with an acceptable safety profile. Further follow-up is necessary to assess the long-term outcomes of TKI treatment in children.

## 1. Introduction

Chronic myeloid leukemia (CML) is a rare myeloproliferative disorder characterized by the presence of Philadelphia (Ph) chromosomes in children and adolescents. Childhood CML accounts for approximately 2–3% of all leukemia cases in children; the age-adjusted incidence rate is 1 per million for children (age 0–14 years) and 2.5 per million in adolescents (age 15–19 years) [1]. Due to its rarity, there is a scarcity of clinical data regarding the management of CML in children and adolescents, and as a result, treatment approaches are currently guided by evidence-based recommendations for adult CML [2,3].

Tyrosine kinase inhibitors (TKIs) are the standard treatment for both adults and children with chronic-phase CML (CML-CP) [4,5]. The advent of newer-generation TKIs, alongside imatinib, has dramatically improved the survival rates of pediatric patients with CML, despite the presence of diverse underlying biological and clinical characteristics in this population [6]. Imatinib was the first front-line TKI to be used and showed comparable efficacy to adult patients [7]. Dasatinib, a second-generation TKI, has been approved for children with CML-CP refractory to front-line imatinib therapy [8]. The results from two phase III studies on adults that compared the efficacy and safety of dasatinib and imatinib showed that dasatinib achieved faster and higher response rates than imatinib in treatment-naïve CML-CP patients, as well as provided clinical benefits to patients who failed to achieve early molecular response (EMR) with front-line imatinib therapy [9,10]. A phase II pediatric trial [11] demonstrated that dasatinib exhibited early and deep molecular responses, consistent with findings from an adult study. However, no large pediatric trials have directly compared these two TKIs.

Acute toxicities associated with TKIs are well described and generally manageable. The long-term use of TKIs in adult CML has been associated with cardiopulmonary, gastrointestinal, and endocrine/metabolic toxicities [12]. However, there is limited knowledge regarding the long-term effects of TKI use in pediatric patients. Prolonged use of imatinib in children, especially when initiated during the prepubertal age, can lead to the dysregulation of bone remodeling, thyroid dysfunction, hypogonadism, and growth disturbances [13,14]. Among these endocrinopathies, growth deceleration has been widely discussed but remains inconclusive. In addition, a recent study reported varying degrees of growth deceleration in pediatric patients treated with dasatinib [15]. Given that the underlying pathophysiology remains unclear, further investigation into endocrine safety profiles is necessary to optimize the management of children receiving TKI treatment.

In this study, we aimed to evaluate the efficacy of dasatinib and imatinib as front-line TKIs and assess adverse events (AEs), including growth rate, associated with dasatinib and imatinib use in children and adolescents with newly diagnosed CML.

## 2. Materials and Methods

### 2.1. Study Cohorts

This single-institution study was conducted at Seoul St. Mary’s Hospital and enrolled pediatric patients (age ≤ 18 years at diagnosis) with Ph chromosome-positive (Ph+) CML-CP or accelerated-phase CML (CML-AP) between 2008 and 2022. The chronic and accelerated phases were defined by the presence of blasts comprising < 10% and 10–19% of the bone marrow cells, respectively. Patients with CML-AP who had matched donors and were initially scheduled for allogeneic hematopoietic stem cell transplantation were excluded. Eligible patients received no previous TKI treatment for CML except hydroxyurea, and all patients except one had an Eastern Cooperative Oncology Group (ECOG) performance status of 0 and adequate hepatic and renal function. The one patient was being managed for nephrotic syndrome at the time of CML diagnosis but had normal renal function. This study was approved by the relevant institutional review board and ethics committee (IRB No. KC23RASI0467).

### 2.2. TKI Treatments

We administered imatinib or dasatinib as front-line TKIs to treatment-naïve pediatric patients with CML in a non-randomized manner. A total of 33 and 18 patients treated with imatinib (IMA group) and dasatinib (DSA group), respectively, were analyzed. Imatinib mesylate (Gleevec, Novartis Pharmaceuticals), supplied as 100 mg capsules, was administered at a dose of 260 mg/m^2^ (rounded to the nearest 100 mg increment) taken once daily with food. Dasatinib (Sprycel, Bristol-Myers Squibb) was administered at a dose of 60 mg/m^2^ once daily. Patients who did not achieve optimal response, experienced disease progression, or demonstrated intolerance to front-line TKI were switched to an alternate TKI.

### 2.3. Evaluation of Efficacy

Patients who underwent front-line or alternate TKI therapy and had at least one efficacy evaluation were assessed for response. Bone marrow cytogenetic responses were measured every 3 months until patients achieved a complete cytogenetic response (CCyR). The CCyR and partial cytogenetic response (PCyR) were defined as 0% and 1–35% Ph-positive cells, respectively, in 400 bone marrow interphase cells (*BCR/ABL* dual color dual fusion translocation probes, Abbott Molecular, USA). Molecular responses were monitored using quantitative reverse-transcriptase polymerase chain reaction (Real-Q *BCR::ABL1* Quantification kit, Biosewoom, Seoul, Korea) every 3 months during the treatment period. MR2, MR3 (major molecular response, MMR), MR4, and MR4.5 are defined as the transcript ratio of *BCR::ABL1/ABL1* being ≤1%, ≤0.1%, ≤0.01%, and ≤0.0032%, respectively, obtained from peripheral blood leukocytes on an international scale (IS) [16]. Deep molecular response (DMR) includes MR4 and MR4.5. CCyR is equivalent to *BCR::ABL1* ≤1% IS or MR2 [4,17]. The response to TKI (classified as optimal, warning, or failure) follows criteria based on the adult European LeukemiaNet (ELN) criteria, which can be reasonably used in children [2,3]. Disease progression was defined as loss of either CCyR or MMR and evolution to CML-AP or the blastic phase at any time.

### 2.4. Safety Analysis

A safety analysis was conducted for all eligible patients who received at least one dose of a TKI. AEs were graded according to the Common Toxicity Criteria Version 5.0. Safety assessment consisted of regular assessment of vital signs; physical condition; laboratory parameters, including hematologic and endocrinologic examinations including Tanner stage; and anthropometric profiles, including height, weight, and body mass index (BMI). Anthropometric profiles were recorded at the initiation of TKI (TKI start), 1 year after TKI start (FU 1), and 1 year after FU 1 (FU 2). All anthropometric data were converted to age- and sex-matched standard deviation scores (SDSs) using a Korean growth chart [18]. Changes in height, weight, and BMI between each assessment point are expressed as delta (Δ). 

### 2.5. Statistical Analysis

Baseline characteristics between the two groups were compared using the independent t-test and Mann–Whitney U test for parametric and non-parametric variables, respectively. Response rates were binomial, and 95% confidence intervals (CIs) were computed using the Clopper–Pearson method. Differences in responses were analyzed using the chi-square or Fisher’s exact test. Time to response and cumulative incidence of response were estimated using the Kaplan–Meier method and compared between the two groups using a two-sided stratified log-rank test. Overall survival (OS) was defined as time from study enrollment to death. Progression-free survival (PFS) was calculated time from the date of starting TKI to the date of documented disease progression. The last follow-up date was 31 March 2023. A *p*-value of <0.05 was considered statistically significant. All statistical analyses were performed using SPSS software (IBM SPSS Statics for Windows, Version 24.0. Armonk, NY, USA: IBM Corp.^®^.

## 3. Results

### 3.1. Patient Characteristics

A total of 51 pediatric patients newly diagnosed with CML-CP (N = 43) or CML-AP (N = 8) were assessed for eligibility. Patients’ baseline characteristics according to study groups are detailed in Table 1. The median age at the initiation of TKI treatment was 12.1 years (range, 2.8–17.6). The median white blood cell count, hemoglobin level, and platelet count were 327.8 10^9^/L (range, 12.9–562.7), 8.4 g/dL (range, 5.4–14.2), and 526.5 10^9^/L (range, 153.0–1767.7), respectively. A total of 37 (72.5%) patients exhibited splenomegaly at diagnosis. No statistically significant differences in baseline characteristics including anthropometric and pubertal profile were observed between the IMA and DSA groups, except for the following: 1) number of female patients, which was higher in the DSA group than in the IMA group (*p* = 0.004), and 2) rate of TKI switching before 12 months, which was higher in the IMA group than in the DSA group (*p =* 0.034) (Table 1). Fifteen (45.4%) of the thirty-three patients in the IMA group switched to an alternate TKI due to intolerance (N = 6), suboptimal response (N = 7), or disease progression (N = 2). Two patients in the DSA group progressed to blastic crisis (BC) (one medullary BC and one isolated central nervous system (CNS) BC) and were treated with systemic and intrathecal chemotherapy, respectively, followed by subsequent allogeneic hematopoietic stem cell transplantation (HSCT). Prior to HSCT, one patient with medullary BC switched to imatinib, while the other patient with isolated CNS BC continued her original TKI, dasatinib [19].

### 3.2. Efficacy

All study patients (N = 51) achieved complete hematologic response at 3 months after treatment. The rate of PCyR at 3 months was similar between the IMA group (76.7%, 95% CI, 57.7–90.1) and DSA group (94.4%, 95% CI, 72.7–99.9) (*p* = 0.110). At 12 months, 92.3% (95% CI, 74.9–99.1) of patients in the IMA group and 100% (95% CI, 75.3–100.0) of patients in the DSA group achieved CCyR (*p* = 0.305) (Figure 1). The rate of EMR, defined as *BCR::ABL1* transcript levels of <10% IS at 3 or 6 months, was significantly higher in the DSA group than in the IMA group (80% vs. 100%, *p* = 0.043). 

The median follow-up durations were 72.1 months (range, 13.3–180.8) in the IMA group and 55.0 months (range, 3.3–139.0) in the DSA group. The rates of MMR at any time during the study period were similar between the IMA (72.7%, 95% CI, 54.5–86.7) and DSA groups (77.8%, 95% CI, 52.4–93.6) (*p* = 0.692). The cumulative rates of molecular response increased over time in both groups, but responses were achieved more rapidly in the DSA group than in the IMA group (Figure 2). At 12 and 24 months, the cumulative rates of MMR were significantly higher in the DSA group than in the IMA group (72.2% vs. 41.2% at 12 months; 100% vs. 68.7% at 24 months, *p* = 0.002) (Figure 2A). In addition, the cumulative rates of DMR at 12, 24, and 60 months were significantly higher in the DSA group than in the IMA group (26.0% vs. 13.8% at 12 months, 43.6% vs. 17.5% at 24 months, and 87.5% vs. 45.4% at 60 months; *p* = 0.004) (Figure 2B).

### 3.3. Efficacy of Dasatinib for Patients with Imatinib Resistance/Intolerance

Of the 33 patients in the IMA group, 14 (42.4%) switched to an alternate TKI (dasatinib in 12 patients and others in 2 patients), with an overall median time to switch of 22.6 months (range, 0.6–106.9). Among the 12 patients who were switched to dasatinib, 4 patients already achieved MR3 at the time of switching. All eight patients whose molecular response was below 0.1% IS at the time of switching had an optimal response to dasatinib, and MR3 was achieved in six of eight patients with a median duration of 9.4 months (range, 3.1–18.4) from the start of an alternate TKI. At 24 months after switching, MR4 was observed in 75.0% (95% CI, 42.8–94.5) of patients treated with dasatinib.

### 3.4. Safety

TKI-related AEs of any grade and grade 3/4 reported in our study cohort are listed in Table 2. Among the patients, 15 (45.4%) and 10 (55.5%) patients in the IMA and DSA groups, respectively, experienced drug-related AEs of any grade during the treatment period. The most common AE of any grade in both groups was myalgia or arthralgia (18% in the IMA group and 22% in the DSA group). Grade 3/4 hematologic toxicity occurred in two (6%) patients in the IMA group and four (18%) patients in the DSA group. One patient in the DSA group experienced dasatinib-related pleural and pericardial effusions, which led to the temporary discontinuation of the drug. The patient resumed dasatinib treatment after the symptoms resolved, and no recurring symptoms were observed.

The median age at the TKI start was 12.1 years, and 47.1% of the patients were in the prepubertal stage. Height, weight, and BMI SDS as well as their changes throughout the follow-up period were compared between the two groups (Figure 3). At the TKI start, FU 1, and FU 2, the height SDSs of patients in the IMA group were 0.14 ± 1.32, −0.27 ± 1.23, and −0.44 ± 1.14, respectively, while in the DSA group were −0.27 ± 0.81, 0.05 ± 1.02, and −0.48 ± 1.14, respectively (Appendix A). Height decline was observed in both groups throughout the follow-up period, with a significantly greater decline in the DSA group than in the IMA group between FU 1 and FU 2 (−0.27 ± 0.24 vs. −0.07 ± 0.23, *p* = 0.035). Both weight and BMI SDS increased from the TKI start to FU 2 in both groups; however, the increment between the two groups was not significantly different (0.55 ± 0.96 vs. 0.53 ± 0.34, *p* = 0.936). The lumbar spine BMD SDS was lower in patients treated with imatinib (−0.19 ± 1.16) than in those treated with dasatinib (−0.08 ± 0.71), but the difference was not significant (*p* = 0.092).

### 3.5. Outcomes

The median follow-up period for the study cohort was 5.4 years (range, 0.4–15.2). Progression to the accelerated or blastic phase occurred in two patients receiving imatinib and two patients receiving dasatinib. The 5-year estimated rates of PFS and OS were similar between the IMA and DSA groups (PFS: 93.7 ± 4.3% vs. 86.6 ± 9.0%, *p* = 0.505; OS: 96.9 ± 3.1% vs. 100%, *p* = 0.540). One patient died of post-transplant complications (grade IV gut and hepatic graft-versus-host disease and sepsis-induced multiple organ failure).

In the study cohort, we attempted to discontinue TKI therapy in seven patients (one patient in the IMA group and six patients in the DSA group) with sustained DMR for at least 24 months. Prior to discontinuation, the median duration of sustained undetected *BCR::ABL1* IS was 60 months. After discontinuation, with a median duration of 36.0 months, three (42.8%, all in the DSA group) of seven patients successfully maintained treatment-free remission during the follow-up period [20]. Four patients who exhibited loss of MMR resumed their original TKI with monthly molecular monitoring, and all patients re-established MR3 within 3 months after TKI resumption. 

## 4. Discussion

This study assessed the efficacy and safety of imatinib (at a dose of 260/ m^2^ once daily) and dasatinib (at a dose of 60 mg/m^2^ once daily) as front-line TKIs in children and adolescent with newly diagnosed CML-CP or CML-AP. The key findings of our study are as follows: (1) both TKIs achieved high CCyR rates by 12 months; (2) dasatinib demonstrated significantly faster and higher rates of EMR, MMR, and DMR than imatinib by 12 and 24 months; and (3) growth decelerations were observed in both TKIs without significant differences. Although there are insufficient data on the comparison of imatinib and dasatinib in children, our findings are consistent with the results of previous studies in adults and provide further evidence of the faster treatment response of dasatinib over imatinib in CML-CP [4,9,10]. Additionally, to the best of our knowledge, this is the first published preliminary report that compares the change in growth velocity between imatinib and dasatinib in children and adolescents with CML. 

The attainment of rapid molecular responses (i.e., achieving *BCR::ABL1* ≤10% at 3 months) has been linked to decreased risk of progression and improved long-term outcomes [21]. Previous studies have demonstrated that a higher proportion of patients treated with second-generation TKIs achieve these treatment goals compared with those treated with imatinib [10,22]. Although the EMR and DMR rates were significantly higher in the DSA group, they did not have an impact on survival outcomes in our study, possibly due to the limited number of cases. This study included a higher proportion of female patients in the DSA group. Given the suggestion of previous reports that the female sex was associated with better PFS and predicted a stable MR4.5 [23,24], this should be considered a potential confounding variable that may impact treatment response. Regarding cytogenetic response, both the IMA and DSA groups achieved excellent CCyR rates at 12 months (92.3% vs. 100%). In addition, among patients who continued front-line TKI therapy throughout the study period, MMR achievement rates at any time were similar between the DSA (77.8%) and IMA groups (72 %), consistent with the achievement rates of previous pediatric pivotal studies [7,11]. However, the observation that 14 (42.4%) patients of the 33 patients in the IMA group required switching to an alternate TKI because of resistance/intolerance is a notable limitation of imatinib.

Dasatinib, as an alternate TKI for patients with CML-CP who developed imatinib resistance/intolerance, is an effective option with clinical benefits. The final 7-year analysis of the phase III CA180-034 study highlighted that dasatinib in the second line demonstrated durable efficacy, the absence of newly noted AEs, and improved PFS and OS in patients achieving EMR [25]. Additionally, dasatinib improved compliance and achieved DMR in patients who were previously intolerant to imatinib [26]. In a phase II pediatric trial, dasatinib showed promising results as an alternative treatment option, inducing CCyR in approximately 50% of patients who were resistant or intolerant to imatinib [11]. Although we observed a limited number of patients (N = 12) who exhibited imatinib resistance/intolerance and switched to dasatinib, it is noteworthy that over 80% of patients achieved MMR within 12 months after switching and subsequently achieved DMR during the study period. Our findings confirm the results of previous studies indicating that dasatinib enables a higher proportion of patients to achieve DMR and that this sustained efficacy offers the potential for more pediatric patients with CML-CP to attain treatment-free remission (TFR). 

The current guidelines for CML allow only adult patients (age ≥ 18 years) with sustained DMR for at least 2 years to be eligible for TKI discontinuation. Recent evidence from a large cohort suggests that TKIs can be safely discontinued in appropriate adult patients without molecular relapse [27,28]. However, the depth and duration of molecular response that needs to be achieved prior to considering TKI discontinuation to attain a long-term TFR remains unclear. Millot et al. reported the outcomes of imatinib discontinuation among 18 pediatric patients with sustained DMR (*BCR::ABL1* ≤0.01%) for at least 24 months, with a TFR rate at 36 months of 56% [29]. More recently, a multicenter prospective study conducted by the Japan Pediatric Leukemia and Lymphoma Study Group reported a TFR rate of 50% at 12 months, and notably, this study demonstrated improved growth velocity and academic performance, which are unique problems in the pediatric population, in pediatric patients who discontinued TKI use before puberty [30]. At our institution, we have also attempted TKI discontinuation for patients treated with either imatinib or dasatinib and who showed sustained DMR (≥MR4.5) for at least 2 years. Of the seven patients, three (42.8%) patients treated with dasatinib maintained TFR for a median duration of 36.7 months after discontinuation.

Both imatinib and dasatinib were well-tolerated by our study patients. Except for one patient treated with dasatinib who experienced grade III pleural and pericardial effusion leading to temporary discontinuation of treatment, none of the cohort patients demonstrated serious organ toxicity, such as pulmonary or cardiovascular disease. Compared with other AEs of TKI use, growth disturbance is a more sensitive issue, particularly in the pediatric population, because longitudinal growth is a unique characteristic that occurs during childhood and adolescence. Previous cohorts that observed statural growth in children treated with imatinib reported height decline, with height SDSs <−1.0, irrespective of sex or pubertal status during the first years of therapy [31,32,33]. The impact of dasatinib on growth velocity has been considered relatively minimal or absent; however, longitudinal growth outcomes after long-term use of dasatinib have rarely been discussed [11,15]. In our patients treated with either imatinib or dasatinib, the height SDS decreased during follow-up, and each height SDS was similar between the two groups. However, the degree of height decline was greater in the DSA group than in the IMA group between FU 1 and FU 2. Our results suggest that dasatinib can alter the height velocity of children at pubertal age as much as or even more evidently than imatinib; therefore, the careful consideration of effects on growth is required when choosing a front-line TKI for pediatric patients. A strength of our study was the direct comparison of height changes between pediatric patients treated with imatinib and dasatinib, as no studies so far have closely examined this issue.

There are several limitations in this study. First, it is a retrospective study conducted at a single institution, resulting in a relatively small sample size. Second, study patients were not randomly assigned to receive either imatinib or dasatinib, resulting in potential selection bias. Consequently, there may be limitations in the statistical power, and the findings should be interpreted with caution. To further validate our findings, a multicenter randomized study with a larger pediatric population is needed. 

## 5. Conclusions

In conclusion, both imatinib and dasatinib are effective and safe front-line TKIs for pediatric patients with CML, providing excellent treatment outcomes in terms of survival. Dasatinib shows faster and higher major and deep molecular responses, with a greater possibility of discontinuing TKI treatment and achieving long-term treatment-free remission in pediatric patients. However, height decline was observed in both imatinib-treated and dasatinib-treated patients. Growth should be monitored closely in pediatric patients who have a high potential for long-term TKI exposure. Further studies in a large cohort are needed to investigate the long-term adverse events and establish the criteria for discontinuation of TKIs in children with CML. 

## Figures and Tables

**Figure 1 cancers-15-03862-f001:**
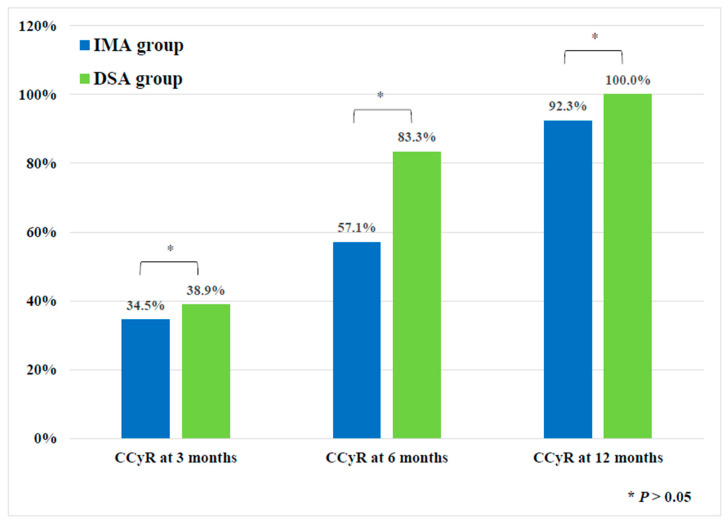
Complete cytogenetic response (CCyR) rates at 3, 6, and 12 months according to study groups.

**Figure 2 cancers-15-03862-f002:**
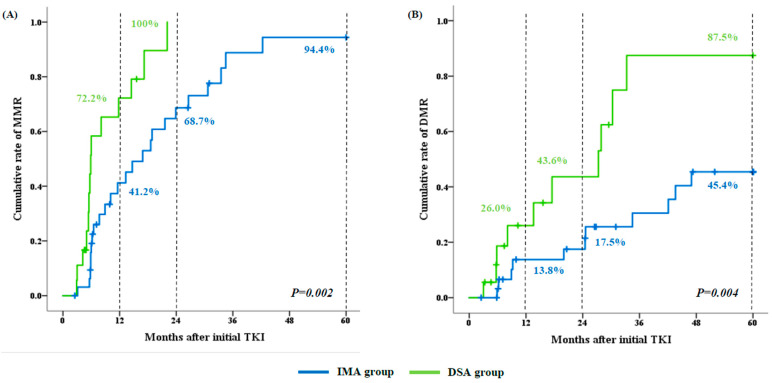
Cumulative rates of molecular response over time according to study groups: (**A**) major molecular response (MMR) and (**B**) deep molecular response (DMR).

**Figure 3 cancers-15-03862-f003:**
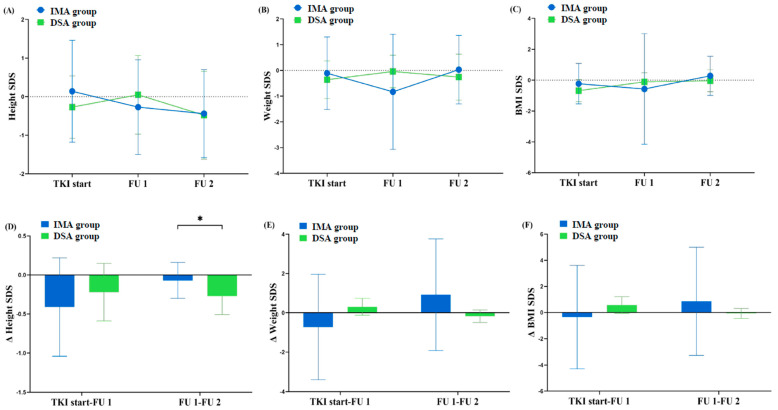
Comparison of height, weight, and body mass index status (**A**–**C**) and their changes in velocity (**D**–**F**) during the follow-up period according to the study groups. Each dot (box) and whisker indicate a mean value and standard deviation, respectively; *p*-values less than 0.05 are indicated with asterisk (*).

**Table 1 cancers-15-03862-t001:** Baseline characteristics of study cohort.

	IMA Group (N = 33)	DSA Group (N = 18)	*p*-Value
Age at diagnosis, median (range)	12.1 (3.5–17.6)	11.8 (2.8–17.5)	-
Sex, N (%) Male Female	23 (69.6) 10 (30.4)	5 (27.7) 13 (72.3)	0.004
CBC at diagnosis, median (range) WBC (10^9^/L) Hemoglobin (g/dL) Platelet (10^9^/L)	293.7 (12.9–562.7) 9.2 (5.7–13.0) 562.0 (162.0–1767.7)	265.7 (20.1–416.2) 8.5 (5.4–14.2) 550.0 (153.0–1558.0)	-
Splenomegaly at diagnosis, N (%) Yes No Unknown	23 (69.6) 6 (18.1) 4 (12.3)	14 (77.8) 4 (22.2) 0	-
*BCR-ABL1* transcript type, N (%) b3a2 b2a2 e14a2 Not applicable	14 (42.4) 8 (24.2) 1 (3.0) 10 (30.4)	12 (66.6) 2 (11.1) 1 (5.5) 3 (16.8)	-
Switching or discontinuation of the initial TKI before 12 months, N (%) Intolerance Suboptimal response Progression	6 (18.1) 7 (21.2) 2 (6.0)	0 0 2 (11.1)	0.034
Tanner stage, N (%) I II III IV V	7 (21.2) 9 (27.2) 8 (24.2) 5 (15.1) 4 (12.1)	4 (22.2) 5 (27.7) 3 (9.0) 2 (11.1) 4 (22.2)	-
Height SDS TKI start FU 1 FU 2	0.14 ± 1.32 −0.27 ± 1.23 −0.44 ± 1.14	−0.27 ± 0.81 0.05 ± 1.02 −0.48 ± 1.14	-
Weight SDS TKI start FU 1 FU 2	−0.11 ± 1.41 −0.83 ± 2.24 0.03 ± 1.33	−0.36 ± 0.73 −0.04 ± 0.63 −0.26 ± 0.89	-
BMI SDS TKI start FU 1 FU 2	−0.23 ± 1.31 −0.57 ± 3.58 0.28 ± 1.27	−0.68 ± 0.71 −0.11 ± 0.59 −0.04 ± 0.71	-

**Table 2 cancers-15-03862-t002:** Any grade and grade 3/4 treatment-related adverse events in the study cohort.

	IMA Group (N = 33)		DSA Group (N = 18)	
Adverse Events	Any Grade	Grade 3/4	Any Grade	Grade 3/4
Hematologic toxicity				
Neutropenia	0	0	3 (16)	3 (16)
Thrombocytopenia	5 (15)	1 (3)	2 (11)	1 (5)
Anemia	1 (3)	1 (3)	1 (5)	0
Extra-hematologic toxicity				
Myalgia or arthralgia	6 (18)	0	4 (22)	0
Nausea and/or vomiting	5 (15)	0	1 (5)	0
Diarrhea	0	0	1 (5)	0
Abdominal pain	2 (6)	0	1 (5)	0
Hematochezia	0	0	1 (5)	0
Superficial/generalized edema	0	0	1 (5)	0
Pleural or pericardial effusion	0	0	1 (5)	1 (5)
Elevated AST or ALT	2 (6)	2 (6)	0	0

Note: values are expressed as no. (%).

## Data Availability

The data presented in this study are available in this article (and Appendix A).

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
