# Peer review of "Front-Line Tyrosine Kinase Inhibitors in Pediatric Chronic Myeloid Leukemia: A Study on Efficacy and Safety"

_cancers, 2023, doi:10.3390/cancers15153862_

Round 1

Reviewer 1 Report

In the manuscript titled “Front-line Tyrosine Kinase Inhibitors in Pediatric Chronic Myeloid Leukemia: A Study on Efficacy and Safety”, the authors evaluate the efficacy and safety of Tyrosine kinase inhibitors (TKI), Dasatinib (DSA) and Imatinib in pediatric CML patients. Following suggestions could be incorporated to improve the quality of the manuscript.

1)    Do all the patient enrolled have Philadelphia positive (Ph) CML? The authors should include it in the text.

2)    Do the patients possess any other comorbities? Can the authors include this in text.

3)    Have the patients ever received any TKI therapy earlier?

4)    The cohort receiving DSA treatment consist of more females than males. Females have in the past shown better responses to TKI treatment. Could the authors shed some light on whether this could be a confounding variable by including this in the discussion.  

5)    The authors mention about puberty being one of the factors influencing the height decrease. Can the authors briefly include information about the puberty status of the patients.

6)    Can the authors include some information about the limitations of this study such as samples size etc. in the discussion.

Author Response

We appreciate the reviewers’ valuable comments regarding our submission. In the attatched file, we have provided point-by-point responses to these comments, and in the revised main manuscript, we have highlighted the updated text. We hope that our revised manuscript now meets the high standards of your journal.

Reviewer 2 Report

Authors described about front-line imatinib or dasatinib in pediatric patients with CML.

1. What treatment did they receive 2 other imatinib intolerance/resistant patients?

2. What treatment did they receive 2 dasatinib failure patients?

Author Response

(The authors gave the same response as above.)
